EMBO
Molecular Medicine

# Soluble CD27 differentially predicts resistance to anti-PD1 alone but not with anti-CTLA-4 in melanoma

Ikuan Sam[1,2,9], Nadine Benhamouda[1,2,9], Lucie Biard[3,10], Laetitia Da Meda [ID][4,10], Kristell Desseaux[3,10], Barouyr Baroudjan[4], Ines Nakouri[4], Marion Renaud[4], Aurélie Sadoux[5], Marina Alkatrib[1], Jean-François Deleuze [ID][6], Maxime Battistella[7], Yimin Shen[6], Matthieu Resche-Rigon[3], Samia Mourah[5,8], Celeste Lebbe [ID][4,11✉] & Eric Tartour [ID][1,2,11✉]

## Abstract

**Metastatic melanoma can be treated with anti-PD-1 monotherapy or in combination with anti-CTLA-4 or anti-Lag3. However, combination therapy is associated with a high risk of toxicity. Recently, we reported that high plasma soluble CD27 (sCD27) levels reflect the intratumoral interaction of CD70-CD27 and dysfunctional T cells in the tumor microenvironment of renal cell carcinoma. In this study, we first characterized the intratumoral expression of CD70 and CD27 in melanoma tumors and their interaction in vivo. We then reported a significant association between baseline sCD27 and anti-PD-1 resistance as assessed by progression-free survival, overall survival, or 12-month complete response in two prospective cohorts of melanoma patients. Multivariate analysis confirmed that sCD27 was independently associated with clinical outcomes. Notably, sCD27 did not predict clinical response to combination therapy in either cohort. This differential predictive value of sCD27 for the two therapeutic options was later confirmed by propensity score analysis. Our results suggest that high plasma sCD27 levels predict poorer efficacy of anti-PD1 monotherapy in metastatic melanoma, justifying therapeutic escalation with a combination of anti-PD1 and anti-CTLA-4.**

**Keywords** CD70-CD27 Interaction; Immunotherapy; Melanoma; Predictive Biomarker; Tumor Microenvironment
**Subject Categories** Biomarkers; Cancer; Skin

## Introduction

Immune checkpoint inhibitors (ICIs) such as anti-PD-(L)1 and anti-CTLA-4 have revolutionized the treatment landscape for patients with advanced melanoma. In the pivotal Checkmate 067 trial, not directly designed to compare nivolumab (anti-PD-1) monotherapy with the nivolumab/ipilimumab combination, the 6.5-year overall survival (OS) rate was 49% in the combination group versus 42% in the monotherapy group (Wolchok et al, 2022). When focusing on melanoma-specific survival (MSS), the 6.5-year rates for the combination and monotherapy groups were 56% and 48%, respectively (Wolchok et al, 2022).

Although the latest ASCO recommendations allow the treatment of patients with metastatic melanoma with anti-PD-1 monotherapy or in combination with anti-CTLA-4 or anti-Lag3 (Seth et al, 2023), the current trend is to treat these patients with the combination therapy with anti-CTLA-4 due to its remarkable long-term outcomes. In addition, the combination of ipilimumab and nivolumab has emerged as the standard of care for treating patients with asymptomatic brain metastases (Long et al, 2018; Tawbi et al, 2021). However, a high risk of toxicity has been reported with grade 3–4 treatment-related adverse events (TRAEs) associated with the combination therapy (59% versus 24% in the nivolumab arm). Forty-two percent of patients receiving the combination discontinued treatment due to TRAEs, compared to only 14% with nivolumab (Weiss et al, 2022).

Although conventional biomarkers of response to immunotherapy (PD-L1, PD-1/PD-L1 interaction, tumor mutational burden (TMB), tertiary lymphoid structure (TLS), CD8[+]T cell infiltration, IFN-γ, and immune activation gene signature) have been reported to be associated with clinical response in patients with metastatic melanoma (Auslander et al, 2018; Cabrita et al, 2020; Gide et al, 2019; Girault et al, 2022; Newell et al, 2022; Voabil et al, 2021),

[1]Universite Paris Cite, INSERM, PARCC, Paris, France. [2]Department of Immunology, APHP, Hôpital Europeen Georges Pompidou (HEGP)-Hôpital Necker, Paris, France. [3]APHP, Department of Biostatistics and Medical Information, APHP, Saint-Louis Hospital, Paris, INSERM, UMR-1153, ECSTRRA Team, Paris, France. [4]Universite Paris Cité, APHP Dermato-Oncology, Cancer Institute AP-HP, Nord Paris Cité, INSERM U976, Saint Louis Hospital Paris, Paris, France. [5]Department of Pharmacology and Tumor Genomics, Hôpital Saint Louis, Assistance Publique-Hôpitaux de Paris, Paris, France. [6]Fondation Jean Dausset-CEPH (Centre d'Etude du Polymorphisme Humain), CEPH-Biobank, Paris, France. [7]Department of Pathology, Hôpital Saint Louis, Assistance Publique-Hôpitaux de Paris, Paris, France. [8]Université Paris Cité, INSERM UMR-S 976, Team 1, Human Immunology Pathophysiology & Immunotherapy (HIPI), Paris, France. [9]These authors contributed equally as first authors: Ikuan Sam, Nadine Benhamouda. [10]These authors contributed equally: Lucie Biard, Laetitia Da Meda, Kristell Desseaux. [11]These authors contributed equally as last authors: Celeste Lebbe, Eric Tartour. ✉E-mail: Celeste.lebbe@aphp.fr; eric.tartour@aphp.fr

none are currently integrated into the clinical management of patients. Presently, no biomarker exists to distinguish patients who would benefit from monotherapy versus those requiring a combination therapy with anti-PD-1 and anti-CTLA4.

CD70, a member of the TNF superfamily and a costimulatory molecule expressed on antigen-presenting cells, interacts with CD27 on T cells. This interaction plays a critical role in their priming, effector functions, differentiation, and memory generation (Denoeud et al, 2011). Our recent findings showed CD70 expression on renal cancer tumor cells, where its chronic interaction with CD27-expressing T cells led to their apoptosis and cleavage of membrane CD27 to a soluble form (sCD27) found in plasma (Benhamouda et al, 2022). Elevated plasma sCD27 levels could reflect the presence of dysfunctional T cells resistant to reactivation by anti-PD-1/PD-L1. Our research indicated that high plasma sCD27 concentrations correlate with resistance to anti-PD-1 immunotherapy, but not to anti-angiogenic therapy in renal cancer patients (Benhamouda et al, 2022).

In the present study, using data from two observational prospective cohorts, we aimed to determine whether the predictive value of sCD27 in anti-PD-1 therapy resistance, as observed previously, extends to melanoma, known to express CD70 (Pich et al, 2016). Considering previous reports showing that anti-CTLA-4 mainly targets lymph node T cells and increases tumor infiltration by new effector T cells (Ribas et al, 2010; Wei et al, 2019), intratumoral T cell dysfunction reflected by sCD27 could be less associated with resistance to the anti-CTLA-4 + anti-PD-1 combination. Unlike monotherapy, which initially targets T cells in the tumor microenvironment (TME), there is a possibility of secondary recruitment of effector T cells with combination therapy (Luoma et al, 2022; Yost et al, 2019).

## Results

### Cells expressing CD27 and CD70 interact in the TME of metastatic melanoma patients

As previously documented in the literature (Pich et al, 2016), our mIF analysis revealed that metastatic melanoma patients express CD70 and are significantly infiltrated by CD27+T lymphocytes (Figs. 1A and 2C). Interactions between CD70 and CD27 in the TME are also observed (Fig. 2A,B,D,E). Unlike findings in renal cell carcinoma (Benhamouda et al, 2022), we found a predominant interaction between CD27 and CD70 expressed by stromal cells compared to tumor cells (Fig. 2B,D,E), even when considering only the interaction with CD8+T cells expressing CD27 (Fig. 2E). Correspondingly, 87% of CD70 expression was found on non-tumoral cells (Fig. 2F,G), while CD27 was partially expressed by CD8+T cells (Fig. 2C). Re-analysis of public scRNAseq data (GSE120575) from Sade-Feldman et al (Sade-Feldman et al, 2018), where CD45+ leukocytes were sorted from 48 melanoma tumor samples showed that CD70 gene expression is higher in some immune cell populations, such as plasma cell (cluster G2), exhausted CD8+T cells or lymphocytes (G6, G9, G11) and regulatory T cells (G7) (Fig. 2H). We also showed that CD27 expression was enriched in exhausted CD8+T cells (Fig. 2H), which is consistent with our previous findings in renal cancer (Benhamouda et al, 2022).

Previous findings from our group showed a positive correlation between the levels of interaction between CD27 and CD70 in the TME and sCD27 concentrations in plasma (Benhamouda et al, 2022), prompting us to measure these concentrations in patients with metastatic melanoma. sCD27 concentrations were higher in

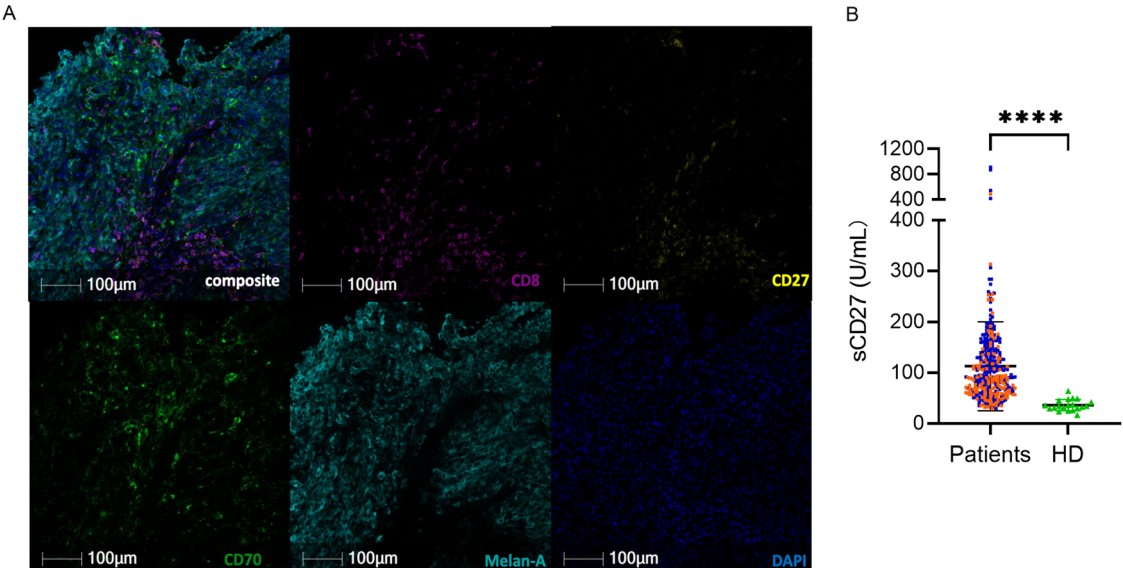

**Figure 1. Melanoma patients express CD70 in their tumors and show an increase in plasma soluble CD27 levels.**

(A) Representative image of multiplex immunostaining performed on formalin-fixed, paraffin-embedded melanoma tissues showing expression of CD8 (purple), CD27 (yellow), CD70 (green), Melan-A (tumor; Cyan). DAPI (blue) was included for cell segmentation. Cell phenotyping was performed using the HALO software, and isotype control antibodies were included in each experiment. (B) Baseline plasma concentrations of sCD27 (U/mL) in metastatic melanoma patients from the PREDIMEL cohort ($n = 138$, orange) and the MelBase cohort ($n = 210$, blue), along with age-matched healthy donors ($n = 20$) as controls. Data are presented in mean ± SD. P values were calculated by using the two-tailed Mann–Whitney test. ****$P < 0.0001$. Source data are available online for this figure.

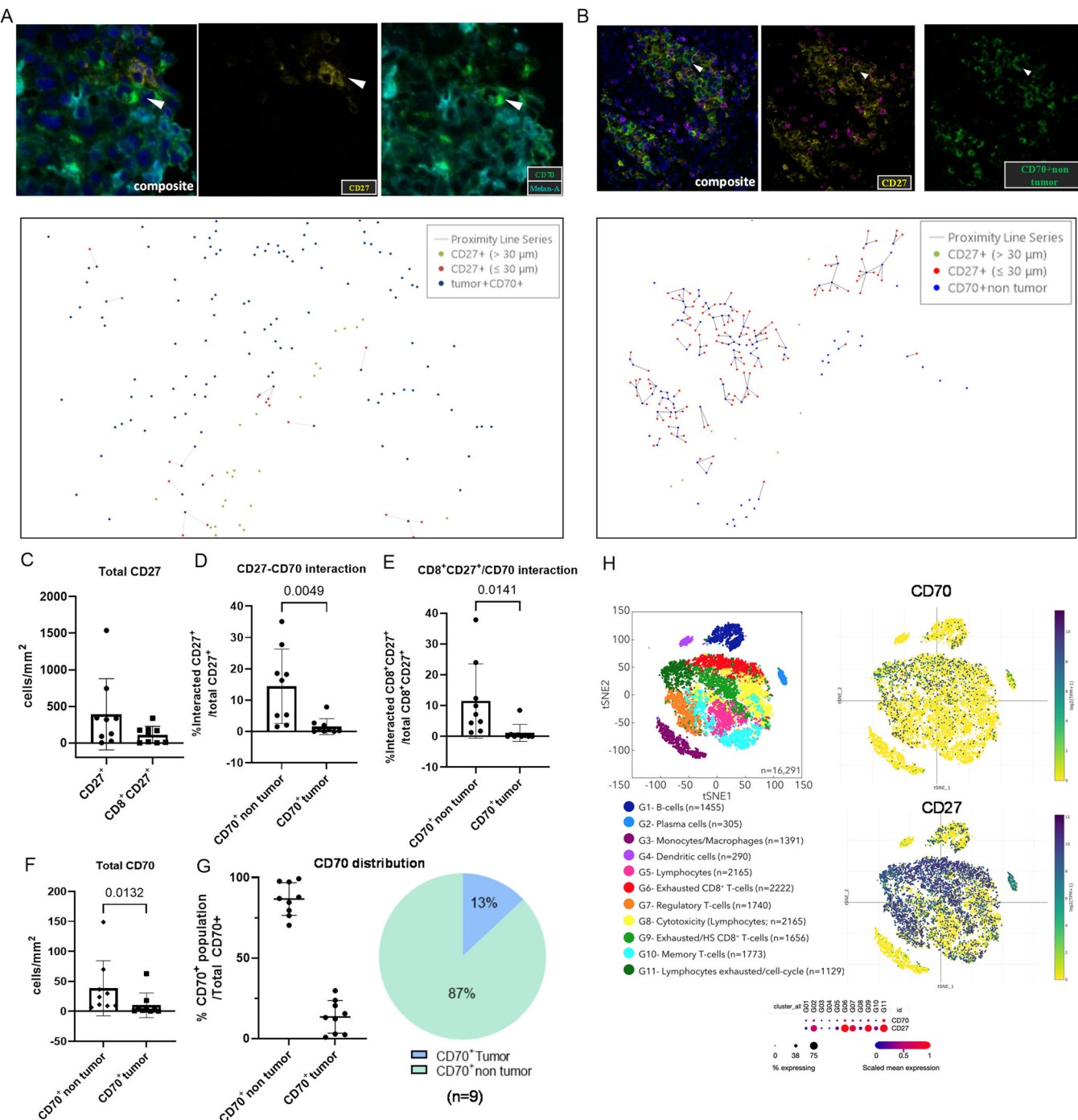

**Figure 2. CD27 interacts with CD70 expressed by tumor and non-tumor cells in the tumor microenvironment of melanoma patients.**

Representative image showing the interaction of CD27-expressing cells with CD70 expressing tumor cells (CD70+Melan-A+) (A, upper) and non-tumor cells (CD70+Melan-A-) (B, upper). Spatial analysis was realized by measuring distance between CD27+ cells and CD70+ tumor cells (A, lower) and CD70+ non-tumor cells (B, lower). Green dots represented CD27+ cells out of 30 μm of CD70+ cells. Red dots represented CD27+ cells within 30 μm of CD70+ cells. Blue dots represented CD70 positive cells. (C) Density of total CD27+ cells and CD8+CD27+ cells in melanoma (n = 9). (D) Percentage of interacted CD27+ with CD70+ non-tumor or tumor cells among total CD27+cells (n = 9). (E) Percentage of interacted CD8+CD27+ with CD70+non-tumor cells or tumor cells among total CD8+CD27+ cell number (n = 9). Density (F) and average distribution (G) of CD70 expression in tumor cells (Melan-A+) and non-tumor cells (Melan-A-) in melanoma (n = 9). (H) Distribution of CD27 and CD70 gene expression in different cell cluster identified by public scRNAseq data analysis in melanoma (GSE120575, Sade-Feldman et al, 2018). tSNE plot of clustered CD45+ immune cells from melanoma tumors after sample normalization and integration in melanoma (left). The normalized log-scale count of CD70 (upper right) and CD27 (lower right) gene expression in the various clusters. Heat map (lower middle) of mean and percentage of CD27 and CD70 expression in each cluster. The data are shown as dots with mean ± standard error of the mean (SEM). Significance was determined by paired t test. Values of P < 0.05 were considered statistically significant. White arrows correspond to the interaction between cells expressing CD27 or CD70. Source data are available online for this figure.

patients with metastatic melanoma (mean ± standard deviation [SD] = 112 ± 86 U/mL) compared to age-matched healthy subjects (mean ± SD = 36 ± 10.9 U/mL; Fig. 1B).

## Elevated baseline sCD27 plasma levels are associated with resistance to anti-PD-1 monotherapy but not to combination therapy with anti-CTLA-4 in the PREDIMEL cohort

In an initial group of patients with metastatic melanoma (PREDIMEL cohort) receiving either anti-PD-1 ($n = 74$) or combination therapy ($n = 64$) as first-line treatment, sCD27 concentrations, evaluated as continuous variables (Fig. 3A) or via an optimal cut-off (100 U/ml), were associated with resistance to anti-PD-1 but not with the combination of anti-PD-1 and anti-CTLA-4 (Figs. 3A and 4A,B, Table EV1). This association was found for both PFS and OS in the anti-PD-1 cohort. In the bi-therapy group, only IL-6 predicts OS (Table EV1), while age, ECOG, LDH and neutrophil/lymphocyte ratio, CRP and IL-6 are correlated with PFS (Table EV1). There was an imbalance for age (younger patients in the bitherapy group) and LDH (higher % of patients with normal LDH in the bitherapy cohort) (Appendix Table S1).

Interestingly, in patients treated with anti-PD-1 alone, elevated sCD27 levels were also associated to the absence of clinical CR at 12 months (Table EV2) or clinical benefits (CR, PR, SD).

Among other factors included in Cox model analysis in PREDIMEL cohort, only AJCC M1c stage and the presence of more than three metastatic sites or brain metastases were associated with PFS in patients treated with anti-PD-1 alone (Fig. 3A). ECOG PS and the presence of more than three metastatic sites correlated with OS in the monotherapy cohorts (Fig. 3A). Other prognostic factors at diagnosis (Breslow, ulceration, lymph nodes, localization, histology) did not predict clinical response to anti-PD-1 (Appendix Table S3). To assess possible biases in identifying sCD27 as a novel biomarker of resistance to anti-PD-1 monotherapy, we investigated whether confounding factors might exist (Fig. 5). We found that sCD27 concentrations differed according to age (above or below 75 years old), and ECOG PS (Fig. 5A). To further document this link between age and sCD27 concentrations, we estimated a Spearman correlation coefficient of 0.29 (95% CI 0.13–0.44; Appendix Fig. S2). In multivariate analysis, sCD27 remained independently associated with PFS in the PREDIMEL cohort ($p = 0.0024$, 95% CI 1.02–1.11), with consistent results in sensitivity analyses for the variable selection procedure. The multivariate analysis could not be performed for OS due to insufficient number of deaths ($n = 15$, Table 1). Other biomarkers sometimes associated with clinical response to anti-PD-1, such as TMB or total CD8$^+$ T cell infiltration in the tumor microenvironment, were analyzed in this cohort of patients treated with anti-PD-1 alone. We found no correlation between these parameters and OS or PFS (Fig. 3A).

## sCD27 concentrations differentially predicted response to monotherapy versus combined therapy in the MelBase validation cohort

Findings from the PREDIMEL cohort were subsequently validated in the MelBase cohort ($n = 210$). Patients treated with anti-PD-1 with sCD27 concentration >100 U/ml were at a higher risk of progression or death, with a median PFS of 3.2 months (95% CI

2.8–6.6) compared to 9.2 months (95% CI 4.1–27.0) in those with lower (≤100 U/ml) sCD27 concentration ($p = 0.039$; Fig. 4D). Similarly, in terms of OS, patients receiving anti-PD-1 with sCD27 concentration >100 U/ml had a median survival of 13.6 months (95% CI 8.6–23.5), whereas median survival was not reached at 60 months from anti-PD-1 initiation in those patients with lower sCD27 concentration (Fig. 4C; $p < 0.001$). These results were consistent when sCD27 was considered as a continuous variable in the Cox test (Fig. 3B). Receiver operating characteristic curve analysis of sCD27 yielded an area under the curve of 0.73. (CI 95% 0.64–0.84) when predicting OS at 1 year. Regarding confounding factors in the MelBase cohort, sCD27 level differed according to age ($p = 0.002$; Spearman correlation coefficient r = 0.27, [95% CI 0.14–0.39]), lactate dehydrogenase (LDH) concentrations ($p = 0.003$), and BRAF mutation ($p = 0.038$; Fig. 5B). As observed in the PREDIMEL cohort, the multivariate analysis, with various factors that were statistically significant in univariate analysis, revealed that sCD27 remained independently associated with PFS and OS at the 5% significance level (Table 1). Sensitivity analyses on the variable selection procedure yielded consistent results for the OS multivariate model selection performed using the MELBASE dataset only.

Consistent with observations in the PREDIMEL cohort (Table EV1), no association was found in the MelBase cohort between sCD27 plasma levels and clinical outcomes (PFS, OS and clinical benefit) in patients treated with anti-PD-1 combined with anti-CTLA-4 (Fig. 4C,D, Table EV3). Indeed, in melanoma patients with low sCD27 (≤100 U/ml) treated by monotherapy, 39/55 patients (71%) from PREDIMEL and 32/42 patients (76%) from MelBase had clinical benefit (CR + PR + SD) while 16/55 (29%) patients from PREDIMEL and 10/42 (24%) patients from MelBase had progressive diseases. In contrast, in patients with high sCD27 (>100 U/ml), only 42% (8/19) patients from PREDIMEL and 47% (22/47) patients from MelBase had clinical benefits, and 58% (11/19) from PREDIMEL and 53% (25/47) from MELBASE progressed. Nevertheless, these results could be explained by an imbalance in the distribution of certain clinical prognostic parameters, as observed for age, the presence of BRAF mutation, more than 3 metastatic sites and AJCC M1 stage at inclusion between patients included in monotherapy or combination therapy in the MelBase cohort (Appendix Table S2). To take this possible bias into account, outcomes between treatment groups were compared using a propensity score approach, revealing that sCD27 significantly discriminated clinical response between these two treatments for OS (interaction $p = 0.006$), clinical benefit (interaction $p = 0.01$), and trends for PFS (interaction $p = 0.09$; Table EV4).

Regarding conventional markers of inflammation (CRP and IL-6), in the 2 cohorts of Predimel and MelBase patients treated with monotherapy, elevated CRP (>5 mg/L) and IL-6 (>10 pg/ml) concentrations correlated with worse OS (Fig. 3A,B). In the cohort of MelBase patients treated with combination therapy, elevated IL-6 and CRP concentrations also correlated with worse OS (Table EV3). These 2 markers appear to be prognostic rather than differential markers of response to anti-PD-1 monotherapy or anti-PD-1 and anti-CTLA-4 combination therapy. These results are in line with the literature, where elevated IL-6 and CRP concentrations are associated with reduced survival in melanoma patients treated with anti-PD-1 monotherapy, or combined with anti-CTLA-4 in dual therapy, or treated with Dacarbazine (chemotherapy) (Laino et al, 2020).

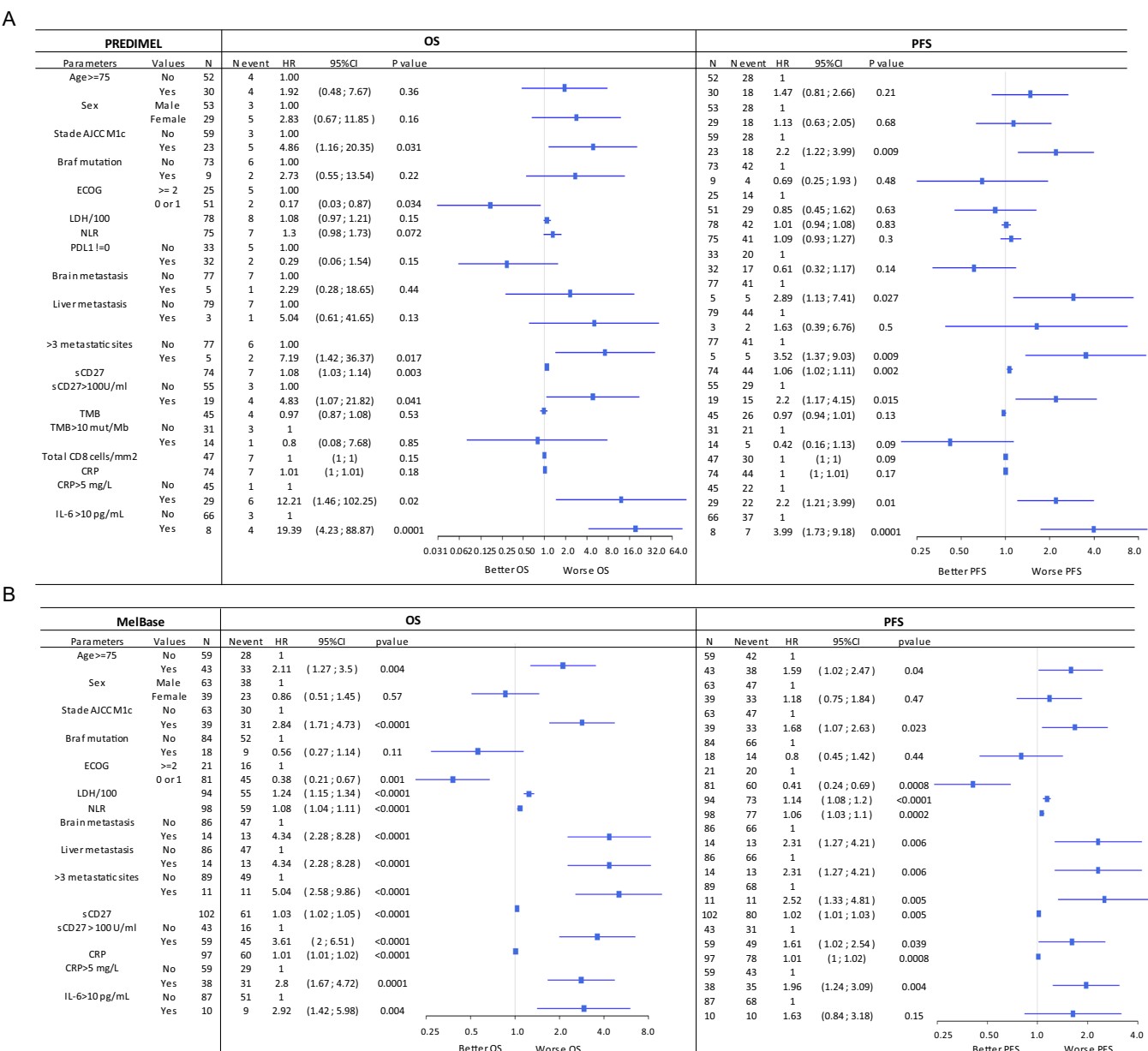

**Figure 3. Clinical and biological variables in the prediction of overall survival (OS) and progression-free survival (PFS) in melanoma patients treated by anti-PD-1 alone in both cohorts.**

Forest plot displaying the univariate Cox's model Hazard Ratios (HRs, blue squares) for OS (**A** left, **B** left) and PFS (**A** right, **B** right) and 95% confidence intervals (CI, solid horizontal blue segments) of baseline clinical and biological variables in the PREDIMEL cohort (**A**) and the MelBase cohort (**B**). Concentration of sCD27 was evaluated either as a continuous variable or dichotomized using a 100 U/ml cut-off. A two-sided $p < 0.05$ was considered significant. N indicates the number of patients in the subgroup defined by the characteristics (e.g., 59 patients aged <75) and N event indicates for each subgroup the number of events among them. The dots and solid lines represent HR and 95% CI. Source data are available online for this figure.

## Discussion

In this study, we evidenced increased levels of soluble CD27 in the plasma of patients with metastatic melanoma across two patient cohorts. Our findings indicate that patients with higher sCD27 levels exhibit poorer response to anti-PD-1 therapy in terms of OS, PFS, and 12-mo CR. Interestingly, these elevated sCD27 levels did not predict response to combination therapy with anti-PD-1 and anti-CTLA-4.

Other predictors of response to immunotherapy such as TMB, CD8[+]T cell infiltration or IFNγ signature data of the literature are not always convergent. For example, regarding the response to anti-PD-1, one study from Johnson et al (Johnson et al, 2016) showed that patients who responded to anti-PD-1/PD-L1 had higher TMB compared to non-responders. However, in another study, TMB failed to predict overall response in anti-PD-1 treated melanoma (Miao et al, 2022). In addition, in melanoma, high TMB was associated with clinical benefit in patients receiving ipilimumab alone

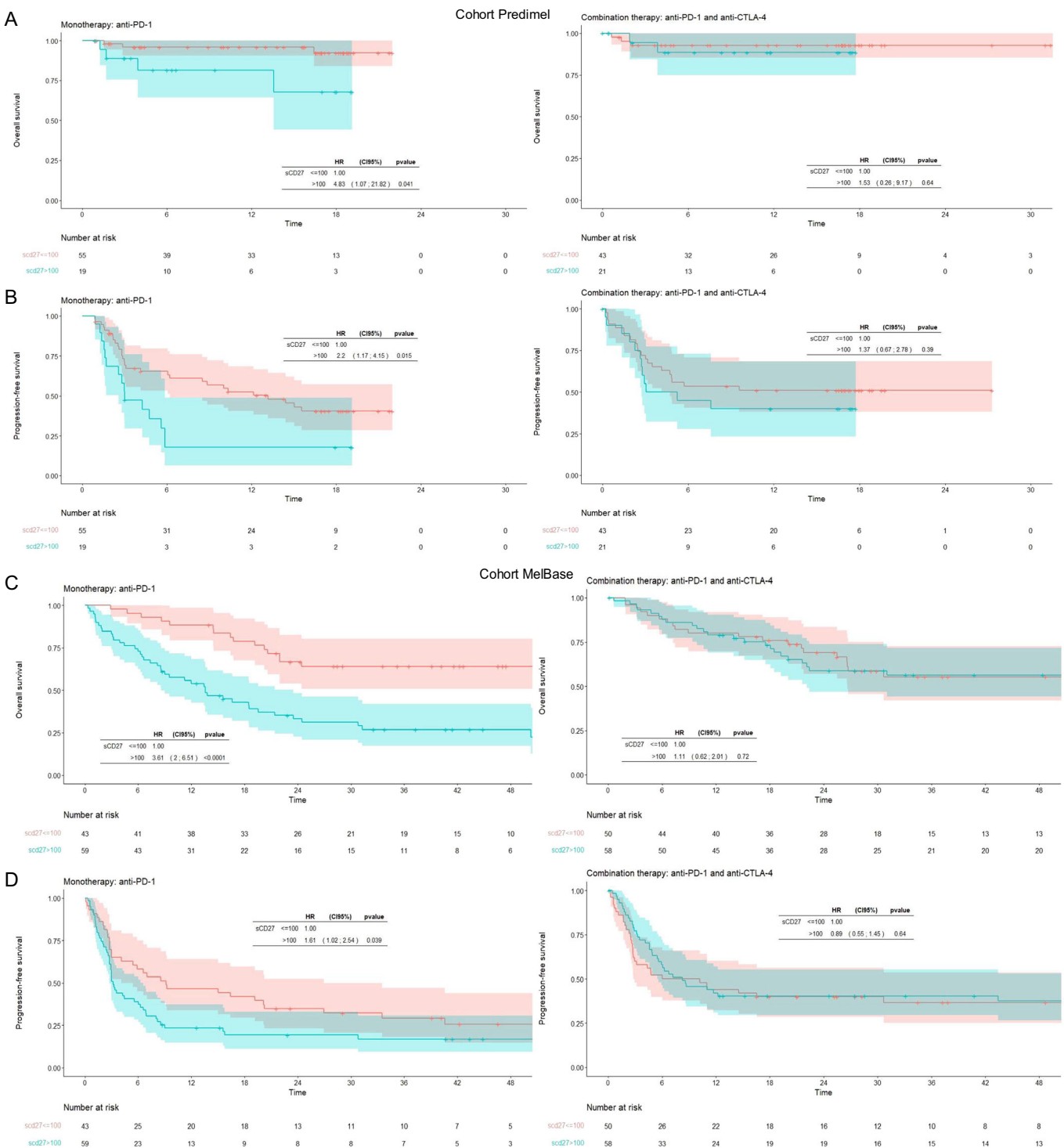

**Figure 4. Concentrations of sCD27 differentially predict overall survival (OS) and progression-free survival (PFS) in two cohorts of metastatic melanoma patients treated with nivolumab or in combination with ipilimumab.**

Metastatic melanoma patients from two cohorts, PREDIMEL (**A**, **B**) and MelBase (**C**, **D**), treated with either anti-PD-1 (**A–D** Left) or a combination of anti-PD-1 and anti-CTLA-4 (**A–D** Right). Kaplan–Meier curves for OS (**A**, **C**) or PFS (**B**, **D**) stratified on high (>100 U/ml) versus low (≤100 U/ml) concentrations of sCD27. Cox's Model Hazards Ratio (HR), 95% confidence intervals, and Wald test *p*-value are presented. Source data are available online for this figure.

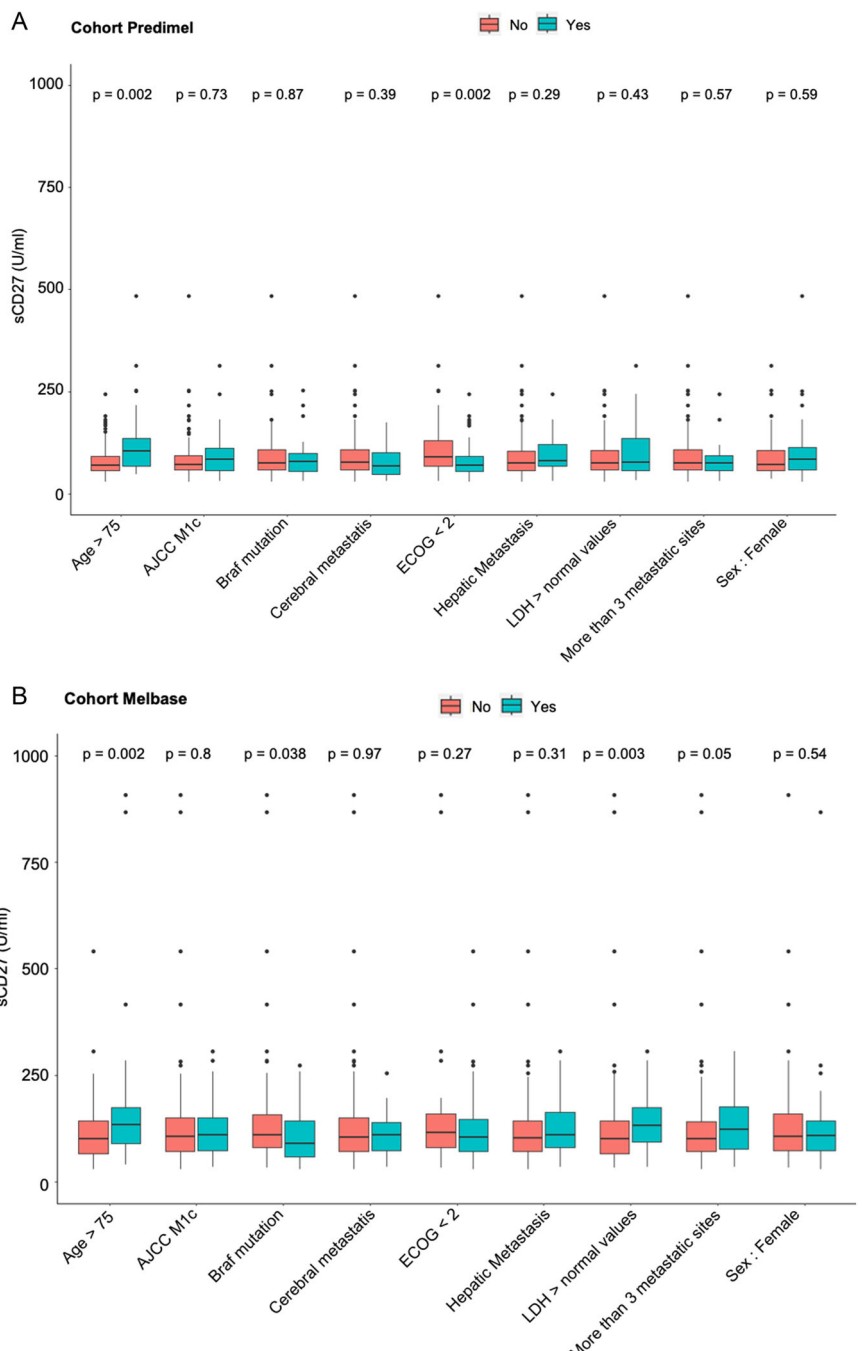

**Figure 5. Association between clinical and biological parameters and sCD27 concentrations.**

Boxplot depicts sCD27 concentrations according to clinical and biological variables in the PREDIMEL cohort ($n = 139$) (**A**) and in the MelBase cohort ($n = 210$) (**B**). A probability value of $p < 0.05$ (Wilcoxon rank sum test) was considered significant. For boxplots, solid black segments represent median values, lower and upper hinges correspond to the first and third quartiles, whiskers extremities extend to the most extreme data point no further than 1.5 times the interquartile range (length of the box) away from the box, and dots correspond to outliers with minima and maxima being the extreme dots. Missing data: PREDIMEL cohort: ECOG $n = 17$, $n = 11$ for LDH; MELBASE cohort: $n = 1$ for BRAF mutation, $n = 4$ for cerebral metastasis, $n = 4$ for liver metastasis, $n = 17$ for LDH, $n = 4$ for numbers of metastatic sites. Source data are available online for this figure.

(anti-CTLA-4) (Snyder et al, 2014) (Van Allen et al, 2015). With respect to the infiltration by total CD8+T cells in the tumor microenvironment, this parameter cannot differentiate the response to anti-PD-1 versus bi-therapy. In some studies, intratumoral total CD8+T cells are associated with prolonged survival in patients treated with nivolumab (Tumeh et al, 2014; Wong et al, 2019) and in those receiving anti-PD-1 plus anti-CTLA-4 (Amaria et al, 2018; Chen et al, 2016). It seems that the impact of CD8 is more pronounced when analyzed in the tumor microenvironment after therapy (Amaria et al, 2018; Chen et al, 2016).

**Table 1.** Multivariate analysis of biological clinical parameters for the prediction of clinical outcome (PFS and OS) in 2 cohorts of melanoma patients with anti-PD-1 alone

| PREDIMEL | | | | | Melbase | | | | |
|---|---|---|---|---|---|---|---|---|---|
| **PFS** | | | | | **PFS** | | | | |
| Parameter | Value | HR | CI95% | P value | Parameter | Value | HR | CI95% | P value |
| sCD27 | | 1.07 | 1.02; 1.11 | 0.0024 | sCD27 | | 1.02 | 1.01; 1.04 | 0.011 |
| AJCC M1c | 0 | | | | AJCC M1c | 0 | 1 | | |
| | 1 | 2.16 | 1.19; 3.91 | 0.011 | | 1 | 1.8 | 1.14; 2.84 | 0.002 |
| | | | | | **OS** | | | | |
| | | | | | sCD27 | | 1.03 | 1.02; 1.05 | <0.0001 |
| | | | | | AJCC M1c | 0 | | | |
| | | | | | | 1 | | | NS |
| | | | | | ECOG | >=2 | 1 | | |
| | | | | | | <2 | 0.55 | 0.29; 1.06 | 0.075 |
| | | | | | Brain metastases | No | 1 | | |
| | | | | | | Yes | 4.14 | 2.09; 8.18 | <0.0001 |
| | | | | | liver metastases | No | 1 | | |
| | | | | | | Yes | 2.03 | 1.02; 4.03 | 0.043 |
| | | | | | Age | | 1.02 | 1; 1.04 | 0.043 |
| | | | | | LDH | | 1.19 | 1.1; 1.29 | <0.0001 |

Step-by-step top-down model selection on Akaike criteria for Predimel PFS and validated on Melbase cohort, OS multivariable model selected only on Melbase due to insufficient death count in Predimel. Missing data on covariates imputed by simple imputation. sCD27, age and LDH were considered as continuous variables.

High baseline IFN-γ signatures were associated with response to (i) anti-PD-(L)1 (Lucas et al, 2023; Gide et al, 2019; Newell et al, 2022; Reschke et al, 2021; Rodig et al, 2018) and (ii) anti-PD-1 and anti-CTLA-4 (Reijers et al, 2023a; Rozeman et al, 2021) (Gide et al, 2019) (Coleman et al, 2023). This correlation was also true in neoadjuvant setting (Reijers et al, 2023a; Rozeman et al; 2021; Blank et al, 2018; Huang et al, 2019; Reijers et al, 2023b) or in adjuvant setting (Long et al, 2023; Versluis et al, 2024). However, in adjuvant setting, an IFNγ gene expression signature higher than the median was prognostic for prolonged relapse-free survival in both patients treated with BRAF inhibitors, anti-PD-1 or placebo group (Dummer et al, 2020; Long et al, 2023; Versluis et al, 2024).

In neoadjuvant trial, the pathologic response was comparable in high IFNγ patients treated by monotherapy or the combination (ipilimumab + nivolumab) (Lucas et al, 2023). However, in patients with low IFNγ signature, the response is higher in the combination (70%) vs the monotherapy (45%).

The IFNγ signature and sCD27 levels may differentially predict response to anti-PD-1 monotherapy or combination therapy. However, there is no consensus on the number of genes to include in this IFNγ signature and, unlike the sCD27 marker, it requires a tumor biopsy.

Several other biomarkers, such as TME enriched for CD16[+] macrophages (Lee et al, 2023), high expression of Lag3 on CD8[+]T cells in the blood (Shen et al, 2021), IL-17 gene expression signature (Varaljai et al, 2023), MHC-class II membrane expression (Rodig et al, 2018), CD4[+]TH1 expansion in the TME on therapy (Franken et al, 2024), and the presence of high endothelial venules (Asrir et al, 2022), have shown correlations with response to

monotherapy versus combination therapy. However, none are currently used for patient management.

A signature based on clinical (such as melanoma primary site, ECOG PS, presence/absence of lung and liver metastases, line of treatment) and biological parameters (including mutation status, LDH, neutrophil-to-lymphocyte ratio) was proposed to distinguish between responses to monotherapy and combination therapy. Notably, some of these markers such as LDH and neutrophil-to-lymphocyte ratio, served as both prognostic and predictive biomarker, as evidenced by their identification in both cohorts (Pires da Silva et al, 2022). However, the clinical decision of whether to initiate monotherapy or combination therapy remains challenging in daily practice, where the risk-benefit balance must be assessed without clear biomarkers to guide these decisions.

Age appeared to correlate with sCD27 levels in both of our patient cohorts and was also associated with resistance to anti-PD-1 therapy. However, even after including age in multivariate analyses, sCD27 remained statistically significant. Interestingly, while age over 75 years has been linked to resistance to anti-PD (L)-1 in some cancer studies, this association does not appear in melanoma (Nie et al, 2021). In renal cancer, we found no relationship between age and sCD27 concentrations (Benhamouda et al, 2022).

The predictive impact of sCD27 on resistance to anti-PD-1 therapy has been previously shown in a small cohort of uveal melanoma (Rossi et al, 2022) and in a series of renal cancer patients (Benhamouda et al, 2022). In this last study, we found a strong interaction between tumor cells expressing CD70 and T lymphocytes expressing CD27. This interaction correlated with plasma sCD27 levels, suggesting that CD27 release occurs during this interaction between CD27 and CD70 (Benhamouda et al, 2022).

Our current study revealed that an interaction between CD27 and CD70 in the TME of melanoma patients, with CD70 primarily expressed by stromal cells in this context. Ongoing research aims to identify the specific cell type responsible for this interaction, drawing parallels to findings in colorectal cancer where fibroblasts were found to primarily express CD70 (Jacobs et al, 2018). A correlation was found between plasma sCD27 levels and CD27-CD70 interaction or combined CD70-CD27 expression intratumorally in renal (Benhamouda et al, 2022) and nasopharyngeal carcinoma (Nagato et al, 2024), respectively, suggesting a tumor origin of plasma-soluble CD27. In melanoma, the origin of sCD27 remains undetermined. We did not have pairs of plasma and tumor from the same patient to perform the same correlation analyses.

Previously, we demonstrated that sCD27 levels reflected a dysfunctional and exhausted state of T cells in the TME, potentially contributing to cancer immunotherapy resistance (Benhamouda et al, 2022). Studies by other groups have reported that chronic interaction between CD27 and CD70 may lead to T cell apoptosis (Wasiuk et al, 2017). Recent work by Honjo's group has shown that elevated plasma levels of sPD-1, sPD-L1, and sCTLA-4 correlate with resistance to anti-PD-1 therapy in NSCLC patients (Hayashi et al, 2024). Similarly to sCD27, high levels of these soluble receptors were strongly correlated with the expression of genes associated with terminally exhausted T cells in tumors (Hayashi et al, 2024). Therefore, sCD27 and other soluble inhibitory receptors may represent a novel class of biomarkers reflecting the status of the TME and the presence of T-cell exhaustion.

Patients treated with anti-PD-1 monotherapy or in combination with anti-CTLA-4 were not stratified for potential confounding factors, which prompted us to validate our results using a propensity score.

The interest of sCD27 has only been demonstrated at baseline but not in the follow-up of patients in a longitudinal study.

Our results suggest that high plasma CD27s levels predict poorer efficacy of anti-PD1 monotherapy in melanoma but not for the combination therapy, supporting the need for therapeutic escalation with the anti-PD1 and anti-CTLA4 combination or other combination (anti-Lag3). Considering the frequent expression of CD70 in various tumors beyond melanoma and renal cell carcinoma, such as nasopharyngeal carcinoma, mesothelioma, and glioblastoma, the predictive value of sCD27 may hold promise for broader application across various cancer types.

# Methods

### Reagents and tools table

| Reagent/Resource | Reference or Source | Identifier or Catalog Number |
|---|---|---|
| **Experimental models** | | |
| Patient plasma | Hospital Saint Louis, Paris | |
| **Recombinant DNA** | | |
| **Antibodies** | | |
| CD8 | Cell Signaling Technology | 70306S |
| CD27 | Abcam | AB131254 |
| CD70 | R&D systems | MAB2738 |
| Melan-A | NovusBio | NBP1-30151 (clone A19-P) |
| HRP Anti-Rabbit | ImmuoReagent | GARHRP-050 |
| HRP Anti-Mouse | ImmuoReagent | GAMHRP-050 |
| **Oligonucleotides and other sequence-based reagents** | | |
| **Chemicals, Enzymes and other reagents** | | |
| CF594 | Biotium | 92174 |
| CF680R | Biotium | 92196 |
| CF430 | Biotium | 96053 |
| CF555 | Biotium | 92214 |
| DAPI | Cell Signaling Technology | 4083S |
| **Software** | | |
| inForm image analysis software | Akoya | |
| HALO | Indica Labs | |
| RStudio | Posit Software | |
| **Other** | | |
| Human CD27 (Soluble) Instant ELISA™ Kit | Thermo Fisher Scientific | BMS286INST |
| Bio-Plex Pro Human Cytokine Screening Standards kit (IL-6) | BIO-RAD | #12007920 |
| CRP48 assay | ALY-C kit | #7P5620 |
| PhenoImager HT | Akoya | |
| LEICA Bond RX automaton | Leica Biosystem | |

## Study design and population

The main eligibility criteria for the present analysis included: advanced melanoma (unresectable stage III or histologically confirmed stage IV per American Joint Committee on Cancer seventh edition [AJCC7]) treated with anti-PD1 alone or in combination with anti-CTLA-4 as first-line therapy. For this study, we used two independent prospective observational non-blinded datasets: PREDIMEL and MelBase with Prof Celeste Lebbe (Hôpital Saint Louis) as principal investigator (PI).

PREDIMEL is a prospective, single-center, observational cohort study in patients with advanced melanoma (unresectable stage III/IV) treated with anti-PD1 alone or in combination with anti-CTLA-4. Its objective is to define and validate an immunological signature predictive of the efficacy of anti-PD1 antibodies alone or in combination with anti-CTLA4 antibodies in the treatment of advanced melanoma. Tumor evaluation response was based on immune Response Evaluation Criteria in Solid Tumor (iRECIST) criteria and performed every 3 months. A biobank (supported by clinical data collection) consisting of blood samples (serum, plasma, and peripheral blood mononuclear cells) and tumor samples was established. A total of 180 patients were included in the PREDIMEL study between December 15, 2016, and February 16, 2023. PREDIMEL is registered with ClinicalTrials.gov (NCT02938728) and was approved by ethics committee CPP Ile de France IV 2016/28 and all patients provided signed, written, informed consent

MelBase is a French, multicenter ($N = 26$ centers), prospective observational cohort study with biobanking designed to prospectively follow-up treatment-naïve patients with stage II or unresectable stage III/IV or neoadjuvant melanoma. The ongoing MelBase study has recruited 3498 patients to date. Clinical and biological data have been collected prospectively since December 28, 2013 through electronic case report form (eCRF). These data include demographics, clinical features of initial melanoma diagnosis, medical history, advanced tumor characteristics, treatment history before initiating immunotherapy, details regarding immunotherapy administrations, response assessments based on RECIST criteria, discontinuation status and reasons, rechallenge, and vital status. Data are updated every 3 months, and then at each change in the line of therapy.

MelBase is registered with ClinicalTrials.gov (NCT02828202) and approved by the French Ethics Committee (CPP Ile-de-France XI, number 12027, 2012), and all patients provided signed, written, informed consent.

For the present analysis, 164 patients from the PREDIMEL study (first-line treatment between December 15, 2016, and February 16, 2023; Appendix Fig. S1A) and 210 patients from the MelBase dataset (enrolled at Saint Louis Hospital by 31 January, 2022; this constituted a validation cohort) (Appendix Fig. S1B) were selected. The median follow-up of patients in the PREDIMEL cohort was 11.3 months, including 78 patients with a follow-up of more than 1 year, while of patients in the MelBase study was 22 months, with 98 patients followed for more than 24 months. Overall, 15 (8/82 patients treated with anti-PD-1 alone and 7/82 patients treated with anti-PD-1 and anti-CTLA-4) and 107 (46/108 patients treated with anti-PD-1 alone and 61/102 patients treated with anti-PD-1 and anti-CTLA-4) deaths, respectively, were reported during follow-up.

All the experiments conformed to the principles set out in the WMA Declaration of Helsinki and the Department of Health and Human Services Belmont Report.

### Multiplex immunofluorescence (mIF) staining

Tumor slides, selected by a pathologist, were deparaffinized using a dewax solution (Leica Biosystem). After the tissue rehydration with three serially diluted ethanol solutions (100%, 75%, 50%) and distilled water, antigen retrieval was performed using Epitope Retrieval Solution 2 (Leica Biosystem). Blocking buffer, devoid of animal-derived proteins (Cell Signaling Technology [CST], Massachusetts, United States), was deposited on the slide, followed by incubation with the primary antibody diluted in SignalStain® Antibody Diluent (CST) for 30 min. Subsequently, incubation with a secondary antibody coupled with horseradish peroxidase (HRP; ImmunoReagents, North Carolina, United States), and then CF® Dye Tyramide (Biotium, California, United States) was added. Slides were then heated to strip the primary and secondary antibodies, washed, and blocked again using blocking solution. This process was repeated until all the markers were labeled. In the final step, slides were stained with DAPI (4′,6-diamidino-2-phenylindole; CST) and mounted using mounting medium Fluoromount-G (Thermofisher). The primary and secondary antibodies used in mIF panels are listed in Appendix Table S4. The IF panel on formalin fixation and paraffin-embedding melanoma tumor tissue is composed of CD8, CD27, CD70, and Melan-A markers. The list of antibodies are listed in Appendix Table S4. The mIF panel, composed of different markers, was developed manually, and the

technique was subsequently transferred to the LEICA Bond RX automaton (Leica Biosystem, Wetzlar, Germany). Isotype IgG was used as the negative control for all antibodies.

### Multispectral imaging, phenotyping and spatial analysis

Multiplex stained slides were scanned at 20x magnification using the Vectra® Polaris™ Automated Quantitative Pathology Imaging system version 2 (Akoya Biosciences, California, United States). Regions of interest were selected and 10 representative images per patient were used for analysis. With single stained slides for each marker, a spectral library containing fluorophores emitting spectral peaks was created within inForm image analysis software version 3.1 (Akoya). This spectral library was then used to separate each multispectral image into its individual components, which allows for the color-based identification of all markers in a single image using inForm software.

For interaction studies, images were exported as component data from inForm to HALO software (Indica labs, New Mexico, United States) for phenotypic and spatial analysis (Indica labs, New Mexico, United States). Highplex FL module in HALO was used for cellular phenotyping. Cells were segmented according to DAPI (nucleus) staining. Phenotyping was realized by setting an appropriate threshold to each marker according to the fluorescent intensity, and the same algorithm was applied to all images for homogeneity. To determine the cellular interaction, spatial analysis was realized based on cellular phenotypes with HALO. Cellular distance is calculated from the center of each cell. We defined CD27 positive cells within 30 μm of CD70 positive cells as "interacted CD27" as previously reported (Benhamouda et al, 2022).

### Tumor mutational burden

The whole exome sequencing (WES) was performed on all patients for both tumor and non-tumor samples. Sequences were aligned to the human genome (hg19) using BWA (Li et al, 2009) (v0.7.15). and we used Picard tools (v2.8.2) to remove PCR duplicates and GATK (v3.7) (McKenna et al, 2010) for local indel realignment and base quality recalibration, as recommended in GATK best practices (DePristo et al, 2011). Somatic variants were identified by Mutect2 from GATK (v3.7) (Cibulskis et al, 2013) and VarScan2 (v2.4.4) (Koboldt et al, 2012) by comparing each tumor sample with its matched non-tumor sample, then we used the filter method by NEXTBioinformatics best practices, and filter out low-frequency variants for min mutation depth covered by ≥10 reads with <5% of variant reads in the non-tumor sample, and we excluded mutations included in COSMIC, 1000G, dbsnp, ExAC databases. In this project we defined TMB by synonymous mutation + non-synonymous mutation. But we analyzed also TMB by only non-synonymous mutation, and proved that there is a very good correlation between them (with Pearson > 0.999), as other published analysis (Bevins et al, 2020).

### Single-cell RNAseq analysis of melanoma

Public scRNAseq dataset (GSE120575) of CD45+ immune cells sorted from 48 melanoma samples (Sade-Feldman et al, 2018) are used for the analysis of gene expression of CD27 and CD70. tSNE plots and heatmaps of gene expression were generated using Single

Cell Portal with documented source code available at https://github.com/broadinstitute/single_cell_portal_core.

## Plasma sCD27 concentration measurement by ELISA

The pre-treatment (baseline) plasma sCD27 concentrations in patients with melanoma and healthy donors were measured with the CD27 (soluble) human Instant ELISA Kit (Thermo Fisher Scientific) according to the manufacturer's instructions. sCD27 levels were measured in duplicate. Data were acquired with an MRX Revelation Microplate Reader (DYNEX Technologies).

## Plasma IL-6 and CRP concentration measurement

Pre-treatment plasma IL-6 concentrations in patients were measured using the Bio-Plex Pro Human Cytokine Screening Standard kit (BIO-RAD) according to the manufacturer's instructions, and data were acquired with Bio-Plex 200 Systems (BIO-RAD). Pre-treatment plasma C-Reactive Protein (CRP) levels were measured using the routine clinical CRP48 assay in ALY-C kit (#7P5620) on the ABBOTT Alinity system at European Hospital Georges Pompidou. Some plasma samples measured for sCD27 concentrations could not be measured for IL-6 and CRP due to insufficient volume or missing samples. Patients with plasma levels of IL-6 > 10 pg /mL or CRP > 5 mg/L are considered to have inflammation.

## Statistical analysis

For descriptive analyses, continuous variables are reported as median value and interquartile range (unless otherwise specified), while categorical variables are presented with counts and percentage. Group comparisons were conducted using Wilcoxon's rank sum test for continuous variables and Fisher's exact test for categorical variables. The primary endpoint was overall survival (OS) defined as the time between the date of inclusion (treatment initiation) and the date of death. Patients who were alive at the last follow-up were censored. Secondary endpoints were progression-free survival (PFS) and 12-month clinical benefit. PFS was defined as the time between the date of inclusion (treatment initiation) and the date of disease progression or death, whichever occurred first. Patients who were alive without progression at the last follow-up were censored. Clinical benefit was defined as complete response (CR), partial response (PR), or stable disease (SD) as measured by RECIST. Survival functions for time-to-event endpoints (OS, PFS) were estimated using the Kaplan-Meier method. Factors associated with these clinical endpoints were evaluated using proportional hazards Cox regression models for OS, PFS, and logistic regression models for a 12-month clinical benefit, reporting hazards ratios (HR) or odds ratio, respectively, with their 95% confidence intervals (CI). Associations were initially assessed using a standard approach, with univariate and adjusted models on the PREDIMEL cohort, followed by the MelBase cohort. Multivariate models were selected by a stepwise procedure (backward elimination primarily; stepwise and forward selection were performed as sensitivity analyses) based on the Akaike criterion, using covariates with an unadjusted $p$ value $\leq 0.10$ in univariate models. Multivariate models were selected on the PREDIMEL dataset and then applied to the

**The paper explained**

**Problem**

Immunotherapy is a new cancer treatment that stimulates the patient's own immune cells to fight the cancer. Patients with metastatic melanoma can be treated with immunotherapy using either an anti-PD-1 antibody alone or in combination with an anti-CTLA-4 antibody. The combination is more toxic than anti-PD-1 monotherapy alone. There are no biomarkers to predict the efficacy of these treatments. Most biomarkers require patients to undergo an invasive biopsy that cannot be repeated. We have previously shown that the biomarker CD27 can be measured in plasma and reflects the state of the immune system in the tumor.

**Results**

We have shown that the tumor of melanoma patients is well infiltrated by CD8[+] T lymphocytes expressing CD27, which interacts with the CD70 molecule. These data suggest that the soluble tumor-derived biomarker CD27 can be measured in these patients. We have shown that high levels of this marker in patients' plasma correlate with resistance to anti-PD-1 therapy, but not to combination anti-PD-1 and anti-CTLA-4 therapy. This marker remained significant in multivariate analyses including other biological and clinical variables. These results were confirmed in 2 patient cohorts with more than 370 melanoma patients.

**Impact**

These results provide evidence to guide the treatment of patients with metastatic melanoma. High levels of soluble CD27 correlate with resistance to anti-PD-1 monotherapy and may justify combination therapy for greater clinical benefit.

MelBase dataset, except for OS given the limiting number of deaths in PREDIMEL ($n = 15$). To further evaluate the predictive nature of sCD27 in relation to treatment effect (anti-PD1 alone versus anti-PD-1 and anti-CTLA-4), a propensity-score inference approach was used, employing inverse probability weighting for average treatments effects in the MelBase cohort, with missing data on covariates handled by simple imputation.

We used a time-dependent approach for the ROC curve estimation, namely a cumulative sensitivity/dynamic specificity ROC curve (with IPCW approach described in (Hung and Chiang, 2010; Kamarudin et al, 2017), considering 12 months as the timepoint of interest in the advanced melanoma setting.

sCD27 concentration was primarily evaluated as a continuous variable. A cutoff value was identified for illustrative purposes, considering a minimum $p$-value approach, guided by existing literature, information on the sCD27 measurements, and simplicity (Altman et al, 1994). Specifically, a minimum $p$-value approach (Altman et al, 1994) in a grid-search manner over a range of cutoff values (by 5-point intervals) was performed, and eventually the simpler value notably in case of exaequos was selected. Given the very limited number of events for the OS endpoint, the 100 U/ml cutoff was identified from PFS in the PREDIMEL cohort, and then applied to the MelBase cohort.

Analyses were performed using R statistical platform. Tests were two-sided, at the 5% significance level for the comparisonwise error rate, given the study focused on sCD27 primarily (Bender et al, 2001).

## Data availability

The WES datasets have not been deposited in a public repository as the original informed consent did not allow for such wide dissemination. Aggregated clinical data, informed consent and statistical analysis plan are available upon request to Celeste.leb-be@aphp.fr or eric.tartour@aphp.fr.

The source data of this paper are collected in the following database record: biostudies:S-SCDT-10_1038-S44321-025-00203-9.

## Peer review information

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

## Acknowledgements

We acknowledge and thank Magdalena Benetkiewicz (ScD) for her editorial assistance (editing a first draft for language and grammar). We thank the histological platform of PARCC (C Lesaffre) and the Biological Resources Center and Tumor Bank Platform of HEGP (BB-0033-00063). This study was funded by Institut National of Cancer (Grant PRTK), Agence Nationale de la Recherche (PC SI 2021) and DIM-ITAC Region Ile de France. This study has been performed in the context of the IdEx Université de Paris Cité (ANR-18-IDEX-0001, France 2030).

## Author contributions

Ikuan Sam: Conceptualization; Formal analysis; Investigation; Methodology; Writing—review and editing. Nadine Benhamouda: Conceptualization; Formal analysis; Investigation; Methodology; Writing—review and editing. Lucie Biard: Data curation; Formal analysis; Investigation; Methodology; Writing—review and editing. Laetitia Da Meda: Resources; Data curation; Formal analysis; Methodology. Kristell Desseaux: Resources; Data curation; Formal analysis; Investigation; Methodology; Writing—original draft. Barouyr Baroudjan: Resources; Formal analysis; Validation; Investigation. Ines Nakouri: Resources; Formal analysis; Validation; Investigation. Marion Renaud: Resources; Formal analysis; Validation; Investigation; Methodology. Aurélie Sadoux: Resources; Formal analysis; Validation; Investigation; Methodology. Marina Alkatrib: Formal analysis; Validation; Investigation; Methodology. Jean-François Deleuze: Formal analysis; Validation; Investigation; Methodology. Maxime Battistella: Resources; Formal analysis; Validation; Investigation; Methodology. Yimin Shen: Formal analysis; Validation; Investigation; Methodology. Matthieu Resche-Rigon: Formal analysis; Validation; Methodology. Samia Mourah: Resources; Formal analysis; Validation; Investigation; Methodology. Celeste Lebbe: Conceptualization; Formal analysis; Supervision; Funding acquisition; Validation; Investigation; Visualization; Writing—review and editing. Eric Tartour: Conceptualization; Formal analysis; Supervision; Funding acquisition; Validation; Investigation; Methodology; Writing—review and editing.

Source data underlying figure panels in this paper may have individual authorship assigned. Where available, figure panel/source data authorship is listed in the following database record: biostudies:S-SCDT-10_1038-S44321-025-00203-9.

## Disclosure and competing interests statement

Celeste Lebbe: Consulting fees: BMS, Pierre Fabre, Sanofi, Novartis, MSD, Amgen, Merck, Serono, Roche, Inflax. Payment of honoraria for lectures, présentations, speakers bureaux: Amgen, BMS, Pierre Fabre, Sanofi, Novartis, MSD, Incyte, Pfizer, Roche. Support for attending meeting and or travel: BMS, MSD, Novartis, Pierre Fabre, Roche, Sanofi. Participation on a data safety monitoring board or advisory board: BMS, Pierre Fabre, Sanofi, Novartis, MSD, Amgen, Merck Serono, Roche, Inflax. Research funding institution: BMS, Roche. Eric Tartour: Research contract: Servier, Oseo Pharma, Imcheck Therapeutics. Payment of honoraria for lectures, présentations, speakers bureaux: Sanofi, BMS, Merck-MSD, Olimpe. Participation of Advisory Board: BMS, Astra-Zeneca, Moderna, Amgen.

