## [Peer Review File · EMBO Molecular Medicine]

Plasma CD27 differentially predicts resistance to anti-PD1 alone but not with anti-CTLA4 in melanoma

Ikuan Sam, Nadine Benhamouda, Lucie Biard, Laetitia Da Meda, Kristell Desseaux, Barouyr Baroudjan, Ines Nakouri, Marion Renaud, Aurelie Sadoux, Marina Alkatrib, Jean-François DELEUZE, Maxime Battistella, Yimin Shen, Matthieu Resche-Rigon, Samia Mourah, Céleste Lebbé, and Eric Tartour

Corresponding author(s): Eric Tartour (eric.tartour@aphp.fr) , Céleste Lebbé (celeste.lebbe@aphp.fr)

Review Timeline:

Submission Date:	26th Oct 24
Editorial Decision:	11th Nov 24
Revision Received:	25th Dec 24
Editorial Decision:	20th Jan 25
Revision Received:	25th Jan 25
Accepted:	10th Feb 25

Editor: Lise Roth

Transaction Report: Please note that the manuscript was transferred from another journal where it was originally reviewed. Since the original reviews are not subject to EMBO's transparent review process policy, they cannot be published.

11th Nov 2024

Dear Prof. Tartour,

Thank you for the submission of your manuscript to EMBO Molecular Medicine, following peer-review at a different journal. Briefly, the initial referees had the following main concerns:

- lack of comparison of sCD27 as a predictor of response/resistance to PD1 with other well-known predictors
- missing information/clarifications
- inadequate statistics.

We have sent your revised manuscript to a single reviewer for evaluation, together with your point-by-point rebuttal letter to the initial referees' concerns. We have now received the enclosed report from this reviewer, who is positive and supportive of publication pending revisions to address two points, related to the Cox model and the ROC curve analysis (see below).

We will therefore welcome the submission of your revised manuscript according to this referee's recommendations.

EMBO Molecular Medicine encourages a single round of revision only and therefore, acceptance or rejection of the manuscript will depend on the completeness of your responses included in the next, final version of the manuscript. For this reason, and to save you from any frustrations in the end, I would strongly advise against returning an incomplete revision.

We require:

4) A .docx formatted letter INCLUDING the reviewers' reports and your detailed point-by-point responses to their comments. As part of the EMBO Press transparent editorial process, the point-by-point response is part of the Review Process File (RPF), which will be published alongside your paper.

5) A complete author checklist, which you can download from our author guidelines (<https://www.embopress.org/page/journal/17574684/authorguide#submissionofrevisions>). Please insert information in the checklist that is also reflected in the manuscript. The completed author checklist will also be part of the RPF.

6) All Materials and Methods need to be described in the main text using our 'Structured Methods' format. According to this format, the Methods section includes a Reagents and Tools Table (listing key reagents, experimental models, software and relevant equipment and including their sources and relevant identifiers) followed by a Methods and Protocols section describing the methods, ideally using a step-by-step protocol format. The aim is to facilitate adoption of the methodologies across labs. Please download and fill our Reagents and Tools Table template (.docx), which you can find in our author guidelines:

<https://www.embopress.org/doi/10.15252/msb.20178071>

7) It is mandatory to include a 'Data Availability' section after the Materials and Methods. Before submitting your revision, primary datasets produced in this study need to be deposited in an appropriate public database, and the accession numbers and database listed under 'Data Availability'. Please remember to provide a reviewer password if the datasets are not yet public (see <https://www.embopress.org/page/journal/17574684/authorguide#dataavailability>).

In case you have no data that requires deposition in a public database, please state so in this section ("This study includes no data deposited in external repositories").

Note that the Data Availability Section is restricted to new primary data that are part of this study.

8) For data quantification: please specify the name of the statistical test used to generate error bars and P values, the number (n) of independent experiments (specify technical or biological replicates) underlying each data point and the test used to calculate p-values in each figure legend. The figure legends should contain a basic description of n, P and the test applied. Graphs must include a description of the bars and the error bars (s.d., s.e.m.). Please provide exact p values.

13) Author contributions: CRediT has replaced the traditional author contributions section because it offers a systematic machine readable author contributions format that allows for more effective research assessment. Please remove the Authors Contributions from the manuscript and use the free text boxes beneath each contributing author's name in our system to add specific details on the author's contribution. More information is available in our guide to authors.

16) As part of the EMBO Publications transparent editorial process initiative (see our Editorial at <http://embomolmed.embopress.org/content/2/9/329>), EMBO Molecular Medicine will publish online a Review Process File (RPF) to accompany accepted manuscripts.

In the event of acceptance, this file will be published in conjunction with your paper and will include the anonymous referee reports, your point-by-point response and all pertinent correspondence relating to the manuscript. Let us know whether you agree with the publication of the RPF and as here, if you want to remove or not any figures from it prior to publication. Please note that the Authors checklist will be published at the end of the RPF.

I look forward to receiving your revised manuscript.

Yours sincerely,

Lise Roth

***** Reviewer's comments *****

Referee #1 (Remarks for Author):

This study presents the utilisation of soluble CD27 as a prospective prognostic biomarker for poor survival outcomes in melanoma patients undergoing PD-1 monotherapy, in addition to its potential as a predictive biomarker for the combination therapy of PD-1 + CTLA-4. The authors addressed the majority of the concerns raised by the reviewers. Therefore, this study has implications for patient stratification in advanced melanoma treatments and may be suitable for publication in EMBO Molecular Medicine, provided that two critical concerns are adequately addressed:

- The primary conclusion of this study is the identification of soluble CD27 as an independent prognostic marker in a multivariate Cox regression with backward stepwise elimination. It would be reassuring to also find sCD27 in a forward stepwise multivariate Cox regression analysis. Moreover, the final covariates included in the multivariate Cox model are largely dependent on one another. It is therefore recommended that the most important and prognostic clinical variables in melanoma, such as tumour thickness (Breslow thickness), tumour stage, lymph node involvement, distant metastasis, tumour location, ulceration and histological subtype, should be considered for multivariate analyses after being significant in the univariate Cox model. In the event that any of the aforementioned prognostic clinical variables are unavailable, this should be explicitly stated in the manuscript, as it represents a significant limitation of the study. This is because the Cox model may not be accurate, and other covariates not reported in the manuscript may prove more effective than sCD27 in predicting survival.
- The Receiver Operating Characteristic (ROC) curve analysis for sCD27 should be conducted as a time-dependent ROC, given that this is a survival analysis.

RESPONSE TO REVIEWERS

The primary conclusion of this study is the identification of soluble CD27 as an independent prognostic marker in a multivariate Cox regression with backward stepwise elimination. It would be reassuring to also find sCD27 in a forward stepwise multivariate Cox regression analysis.

Moreover, the final covariates included in the multivariate Cox model are largely dependent on one another. It is therefore recommended that the most important and prognostic clinical variables in melanoma, such as tumour thickness (Breslow thickness), tumour stage, lymph node involvement, distant metastasis, tumour location, ulceration and histological subtype, should be considered for multivariate analyses after being significant in the univariate Cox model. In the event that any of the aforementioned prognostic clinical variables are unavailable, this should be explicitly stated in the manuscript, as it represents a significant limitation of the study. This is because the Cox model may not be accurate, and other covariates not reported in the manuscript may prove more effective than sCD27 in predicting survival.

We agree with the reviewer on these issues. In the two cohorts, we did consider known prognostic factors in melanoma in the analysis of patients' outcomes relevant in the setting of the study population (stage IV or III ineligible to surgery): Also, given the limited sample size and counts of events in the datasets, we used a parsimonious approach to avoid overparameterization of the models (univariate and multivariable). Namely, we considered:

- tumor stage: as stage AJCC IV M1c vs. other (III, IV M1a-b)
- distant metastasis: is directly related to tumor stage above; we also included liver metastasis yes/no and cerebral metastasis yes/no.

Moreover, as suggested by the reviewer, we have further considered the following factors, when available (namely in the PREDIMEL dataset only). Specifically

- tumor thickness (Breslow thickness) at diagnosis
- ulceration at diagnosis
- lymph node involvement (N0 vs N+) at diagnosis
- tumor localization, defined considering light exposure, consistently with current practice in melanoma: [mucosae/palms and foot sole/nails] vs other skin
- histological subtype, which is strongly correlated to tumor location: [mucosal / acral lentiginous] vs others

As such, we have added these known factors in univariate analysis for PFS and OS, using the PREDIMEL dataset. None of the added variables was associated with a p-value < 0.1 in the univariate analysis of PFS. We did not include them in the candidate set for the multivariable analysis, primarily (we included them in a sensitivity analysis, see below). As detailed in the manuscript, there was too few death events to perform a multivariable analysis in OS using the PREDIMEL dataset.

This is reported in the table below for the review and in Table S3 in the revised manuscript. ew death events to perform a multivariable analysis in OS using the PREDIMEL dataset.

Variable	Values	Progression free survival - Univariate analysis					Overall survival - Univariate analysis				
		N	Nevent	HR	95%CI	P value	N	Nevent	HR	95%CI	P value
Breslow		82	46	0.98	0.91-1.06	0.57	82	8	0.97	0.80-1.18	0.79
Ulceration	No	34	19	1			34	3	1		
	Yes	40	22	1.14	0.61-2.11	0.68	40	4	1.37	0.31-6.14	0.68
	Unknown	8	5	1.46	0.54-3.94	0.46	8	1	1.94	0.20-18.9	0.57
Lymph nodes	N0	60	33	1			60	4	1		
	N+	22	13	1.12	0.59-2.13	0.73	22	4	2.76	0.69-11.1	0.15
Localization	Mucosae/Palms/Soles/Nails	7	5	1			7	1	1		
	Other skin	75	41	0.72	0.28-1.82	0.48	75	7	0.67	0.08-5.46	0.71
Histology	Other	77	44	1			77	8			
	Mucosal-acral lentiginous	5	2	0.71	0.17-2.93	0.64	5	0	n/a	n/a	0.50*

* p-value of log-rank test, due to absence of death events in one group

Regarding the selection procedure, we used a backward selection approach primarily, on Akaike's information criterion (AIC). As per the reviewer's suggestion, we also performed forward and stepwise selection on AIC as sensitivity analyses. For the PFS analysis using the PREDIMEL anti-PD1 dataset, sCD27 remained independently associated with PFS in the revised multivariable analysis, whichever the selection procedure (backward and stepwise selection yielded the same model in terms of selected variables, including sCD27 (p=0.002) and AJCC stage M1c (p=0.011); forward selection additionally included cerebral metastasis (p=0.26) using AIC-based selection.

Similarly, for the OS analysis using the MELBASE anti-PD1 dataset, sCD27 remained independently associated with OS in the revised multivariable analysis, whichever the selection procedure (backward and stepwise selection yielded the same model in terms of selected variables, including sCD27 (p<0.0001), cerebral metastases (p<0.0001), LDH level (p<0.0001), liver metastases (p=0.043), age (p=0.043) and ECOG scale (p=0.075); forward selection included the same variables plus the neutrophils/lymphocytes ratio (p=0.20)).

As a sensitivity analysis, we also forced the additional prognostic factors listed above in the candidate set for multivariable analysis for PFS using the PREDIMEL dataset despite their p-value greater than 0.1 in univariate analysis. We found consistent results with the main analysis: sCD27 remained independently associated with PFS, whichever the selection procedure (backward and stepwise selection yielded the same model in terms of selected variables, including sCD27 (p=0.002) and AJCC stage M1c (p=0.011); forward selection included additional variables (cerebral metastasis (p=0.24), Breslow at diagnosis (p>0.99), ulceration at diagnosis (p=0.91), lymph node involvement (p=0.93), localization (p=0.38), histological subtype (p=0.43)) although this latter model should be interpreted with caution given the limited number of events for this analysis (42 events)).

These additional results have been added in the revised manuscript as follows :

- Patients and Methods section: Page 18 line 9-12 "Multivariable models were selected by a stepwise procedure (backward elimination primarily; stepwise and forward selection were performed as sensitivity analyses) based on the Akaike criterion, using covariates with an unadjusted p value ≤ 0.10 in univariate models."
- Results section:
 - o Univariate estimates for added prognostic factors on PFS and OS were added to Appendix Table S3. The following sentence was also added Page 7 : Line 22-24 : Other prognostic factors at diagnosis (Breslow, ulceration, lymph nodes, localization, histology) did not predict clinical response to anti-PD-1 (Table S3).

- Page 8 : line 2-5 “In multivariable analysis, sCD27 remained independently associated with PFS in the PREDIMEL cohort (p=0.0024, 95% CI 1.02-1.11), with consistent results in sensitivity analyses for the variable selection procedure.”
- Page 9 : line 3-8 : “As observed in the PREDIMEL cohort, the multivariable analysis, with various factors that were statistically significant in univariate analysis, revealed that sCD27 remained independently associated with PFS and OS at the 5% significance level (Table 1). Sensitivity analyses on the variable selection procedure yielded consistent results for the OS multivariable model selection performed using the MELBASE dataset only.”

The Receiver Operating Characteristic (ROC) curve analysis for sCD27 should be conducted as a time-dependent ROC, given that this is a survival analysis.

We agree with the reviewer. Indeed, we used a time-dependent approach for the ROC curve estimation, namely a cumulative sensitivity /dynamic specificity ROC curve (with IPCW approach described in [Hung, H. and Chiang, C. Estimation methods for time-dependent AUC with survival data. *Canadian Journal of Statistics*. 2010;38(1):8-26; Kamarudin AN, Cox T, Kolamunnage-Dona R. Time-dependent ROC curve analysis in medical research: current methods and applications. *BMC Med Res Methodol*. 2017 Apr 7;17(1):53]), considering 12 months as the timepoint of interest in the advanced melanoma setting.

20th Jan 2025

Dear Prof. Tartour,

Thank you for submitting your revised study, and please accept my apologies for the delay in getting back to you during this busy time of the year. We have now received the report from the referee who evaluated your revised manuscript. As you will see below, he/she is satisfied with the revisions, and I will therefore be able to accept your manuscript once the following editorial issues are addressed:

1/ Referee's comments:

Please address the remaining comments from the referee.

2/ Manuscript text:

- Please accept previous changes and only keep in track changes mode any new modification.
- Please remove the headings "Limitations" and "Conclusions" in the discussion section.
- Patients and Methods:
 - o Please rename this section "Methods"
 - o Please include the methods currently in the Appendix file in the main manuscript file.
 - o Please provide a statement that the experiments conformed to the principles set out in the WMA Declaration of Helsinki and the Department of Health and Human Services Belmont Report.
 - o Statistics: please provide a statement on randomization procedure, and adjust the checklist accordingly.
 - o Antibodies: please provide dilutions/concentrations.
- Data Availability: this section should be placed before Acknowledgements. Please note that the Data Availability Section is restricted to new primary data that are part of this study, therefore please remove "We could share aggregated data but not individual data. We can provide informed consent and statistical analysis plan. Biological resources could also be available after approval of a proposal to be sent to celeste.lebbe@aphp.fr or eric.tartour@aphp.fr"
- Acknowledgements: Please remove "and funding" from the heading.
- Please remove "Expanded View Figure Legends".

3/ Figures and Appendix:

- EV tables: please upload as individual files; the legends need to be removed from the manuscript and provided in each file.
- Table 1 is not needed as a separate file as it is already provided in the manuscript.
- Appendix file: please add page numbers in the table of content. Please update the nomenclature to Appendix Figure S1, Appendix Figure S2; Appendix Table S1, Appendix Table S2, Appendix Table S3; Appendix Figure S1 has two pages which is fine, however, the legend should be provided after both parts of the figure
- Please address the queries from our copy editors in the figure legends:
 1. Please note that for the figures 1B, p-values and statistical tests are indicated in the legends. However, comparison for the same, "****/****/**/****" has not been represented in the figures. Please rectify this in the figures or legends as applicable.
 2. Please note that the box plots need to be defined in terms of minima, maxima, centre, bounds of box and whiskers, and percentile in the legends of figures 5A, B.
 3. Please note that information related to n is missing in the legends of figures 2D-F; 5A, B.
 4. Please note that the error bars are not defined in the legends of figures 1B.
 5. Please note that the measure of center for the error bars needs to be defined in the legends of figures 3A, B."
 6. Please note that the white arrow heads are not defined in the legend of figures 2A, B. This needs to be rectified.

4/ Source Data:

Please group all source data related to 1 figure into 1 individual folder, subdivided into subfolders for different figure panels. Each document should be clearly labeled with the figure and panel number/letter.

5/ Please note that all corresponding authors are required to supply an ORCID ID for their name upon submission of a revised manuscript. An ORCID identifier is currently missing for C. Lebbe.

6/ Synopsis image: Thank you for providing a nice synopsis image. Please resize it to 550 px wide to 300-600 px high and make sure that the text remains legible. Please note that a cropped portion of your synopsis will be used as thumbnail on our website.

7/ As part of the EMBO Publications transparent editorial process initiative (see our Editorial at <http://embomolmed.embopress.org/content/2/9/329>), EMBO Molecular Medicine will publish online a Review Process File (RPF) to accompany accepted manuscripts.

This file will be published in conjunction with your paper and will include the anonymous referee reports, your point-by-point response and all pertinent correspondence relating to the manuscript. Let us know whether you agree with the publication of the RPF and as here, if you want to remove or not any figures from it prior to publication.

I look forward to receiving your revised manuscript.

Yours sincerely,

Lise Roth

Lise Roth, PhD

Senior Editor

EMBO Molecular Medicine

***** Reviewer's comments *****

Referee #1 (Comments on Novelty/Model System for Author):

The technical quality, novelty, medical impact and appropriateness of the model system in this manuscript are excellent.

Referee #1 (Remarks for Author):

The authors have addressed the main concerns raised by this reviewer. Therefore, the manuscript is suitable for publication in EMBO Mol. Med. after including the methodology and references for the receiver operating characteristic (ROC) curve analysis, which was only described in the point-by-point response. Finally, when referring to Cox regression, the term 'multivariate' is more appropriate than 'multivariable'.

RESPONSE TO EDITORIAL AND REFEREE'S COMMENTS

1/ Manuscript text:

- Please accept previous changes and only keep in track changes mode any new modification.

We have followed your recommendations

- Please remove the headings "Limitations" and "Conclusions" in the discussion section.

We have removed the headings « Limitations » and « Conclusions » in the discussion section

- Patients and Methods:

o Please rename this section "Methods"

We have renamed this section "Methods"

o Please include the methods currently in the Appendix file in the main manuscript file.

We have included the methods previously in the Appendix file in the main manuscript file.

o Please provide a statement that the experiments conformed to the principles set out in the WMA Declaration of Helsinki and the Department of Health and Human Services Belmont Report.

Page 16 Line 18-19 : We have added that « All the experiments conformed to the principles set out in the WMA Declaration of Helsinki and the Department of Health and Human Services Belmont Report ».

o Statistics: please provide a statement on randomization procedure, and adjust the checklist accordingly.

Page 5 line 15 Page 15 line 3 and 5 and 18 : We clarified that PREDIMEL and MelBase were two prospective observational cohorts and adjusted the checklist accordingly.

o Antibodies: please provide dilutions/concentrations.

The list of antibodies and their dilutions/concentrations are listed in Appendix Supplementary Table 4

- Data Availability: this section should be placed before Acknowledgements. Please note that the Data Availability Section is restricted to new primary data that are part of this study, therefore please remove "We could share aggregated data but not individual data. We can provide informed consent and statistical analysis plan. Biological resources could also be available after approval of a proposal to be sent to celeste.lebbe@aphp.fr or eric.tartour@aphp.fr"

We have moved this section before Acknowledgements and removed the sentence « We could share aggregated data but not individual data. We can provide informed consent and statistical analysis plan. Biological resources could also be available after approval of a proposal to be sent to celeste.lebbe@aphp.fr or eric.tartour@aphp.fr ».

- Acknowledgements: Please remove "and funding" from the heading.

We have removed « and funding »

- Please remove "Expanded View Figure Legends".

We have removed « Expanded View Figure Legends ».

3/ Figures and Appendix:

- EV tables: please upload as individual files; the legends need to be removed from the manuscript and provided in each file.

We have uploaded the EV tables as a individual file. The legends have been attached to each table and removed from the manuscript text.

- Table 1 is not needed as a separate file as it is already provided in the manuscript.

We have left Table 1 as a separate file as it is not already included in the manuscript

- Appendix file: please add page numbers in the table of content. Please update the nomenclature to Appendix Figure S1, Appendix Figure S2; Appendix Table S1, Appendix Table S2, Appendix Table S3;

We have added page numbers in the Table of Content and updated the nomenclature for supplementary figures and tables.

Appendix Figure S1 has two pages which is fine, however, the legend should be provided after both parts of the figure

The legend has been moved after both parts of the figure.

- Please address the queries from our copy editors in the figure legends:

1. Please note that for the figures 1B, p-values and statistical tests are indicated in

the legends. However, comparison for the same, "****/**/**/" has not been represented in the figures. Please rectify this in the figures or legends as applicable. We have indicated the correspondence between the "****/**/**/" symbol and the p value in the legend of Figure 1B

2. Please note that the box plots need to be defined in terms of minima, maxima, centre, bounds of box and whiskers, and percentile in the legends of figures 5A, B. In the legends of Fig 5A, B, we have better defined the boxplots : "For box plots, solid black segments represent median values, lower and upper bounds correspond to the first and third quartiles, whisker extremities extend to the most extreme data point no further than 1.5 times the interquartile range (length of the box) away from the box, and dots correspond to outliers with minima and maxima being the extreme dots.

3. Please note that information related to n is missing in the legends of figures 2D-F; 5A, B.

The n (number of subjects) has been added to the legends of Figure 2D-E and 5-A-B

4. Please note that the error bars are not defined in the legends of figures 1B. It has been indicated in the legend of Figure 1B that « Data are presented in mean±SD»

5. Please note that the measure of center for the error bars needs to be defined in the legends of figures 3A, B."

The measure of center for the error bars have been added in the legends of figures 3A, B.

6. Please note that the white arrow heads are not defined in the legend of figures 2A, B. This needs to be rectified.

We have clarified the significance of the white arrow in the legend of Figure 2A, B and added the sentence « White arrows correspond to cells co-expressing CD27 and CD70».

4/ Source Data:

Please group all source data related to 1 figure into 1 individual folder, subdivided into subfolders for different figure panels. Each document should be clearly labeled with the figure and panel number/letter.

5/ Please note that all corresponding authors are required to supply an ORCID ID for their name upon submission of a revised manuscript. An ORCID identifier is currently missing for C. Lebbe.

The Orcid number for C Lebbe is : 0000-0002-5854-7290

:

6/ Synopsis image: Thank you for providing a nice synopsis image. Please resize it to 550 px wide to 300-600 px high and make sure that the text remains legible. Please note that a cropped portion of you synopsis will be used as thumbnail on our website.

We have resized the graphical/synopsis image to 550 px wide to 300-600 px high.

The text remains eligible.

7/ As part of the EMBO Publications transparent editorial process initiative (see our Editorial at <http://embomolmed.embopress.org/content/2/9/329>), EMBO Molecular Medicine will publish online a Review Process File (RPF) to accompany accepted manuscripts.

This file will be published in conjunction with your paper and will include the anonymous referee reports, your point-by-point response and all pertinent correspondence relating to the manuscript. Let us know whether you agree with the publication of the RPF and as here, if you want to remove or not any figures from it prior to publication. Please note that the Authors checklist will be published at the end of the RPF.

We agree with the publication of all the online review process. The author checklist has been modified to include information about the non-randomization procedure.

***** Reviewer's comments *****

Referee #1 (Comments on Novelty/Model System for Author):

The technical quality, novelty, medical impact and appropriateness of the model system in this manuscript are excellent.

Referee #1 (Remarks for Author):

The authors have addressed the main concerns raised by this reviewer. Therefore, the manuscript is suitable for publication in EMBO Mol. Med. after including the methodology and references for the receiver operating characteristic (ROC) curve analysis, which was only described in the point-by-point response. Finally, when referring to Cox regression, the term 'multivariate' is more appropriate than 'multivariable'.

Thank you for your positive review of this manuscript

We have added the the methodology and references for the receiver operating characteristic (ROC) curve analysis in the Statistical section.

The term « multivariable » has been replaced by « multivariate ».

10th Feb 2025

Dear Prof. Tartour,

Thank you for bearing with the last editorial matters. I am pleased to inform you that your manuscript is accepted for publication and is now being sent to our publisher to be included in the next available issue of EMBO Molecular Medicine!

If you have any questions, please do not hesitate to contact the Editorial Office.

Thank you for your contribution to EMBO Molecular Medicine.

With kind regards,

Lise
